# Unusual competition of superconductivity and charge-density-wave state in a compressed topological kagome metal

F. H. Yu[1], D. H. Ma[1], W. Z. Zhuo[1], S. Q. Liu[1], X. K. Wen[1], B. Lei[1], J. J. Ying[1 ✉] & X. H. Chen [1,2,3 ✉]

Understanding the competition between superconductivity and other ordered states (such as antiferromagnetic or charge-density-wave (CDW) state) is a central issue in condensed matter physics. The recently discovered layered kagome metal $AV_3Sb_5$ ($A$ = K, Rb, and Cs) provides us a new playground to study the interplay of superconductivity and CDW state by involving nontrivial topology of band structures. Here, we conduct high-pressure electrical transport and magnetic susceptibility measurements to study $CsV_3Sb_5$ with the highest $T_c$ of 2.7 K in $AV_3Sb_5$ family. While the CDW transition is monotonically suppressed by pressure, superconductivity is enhanced with increasing pressure up to P1 ≈ 0.7 GPa, then an unexpected suppression on superconductivity happens until pressure around 1.1 GPa, after that, $T_c$ is enhanced with increasing pressure again. The CDW is completely suppressed at a critical pressure P2 ≈ 2 GPa together with a maximum $T_c$ of about 8 K. In contrast to a common dome-like behavior, the pressure-dependent $T_c$ shows an unexpected double-peak behavior. The unusual suppression of $T_c$ at P1 is concomitant with the rapidly damping of quantum oscillations, sudden enhancement of the residual resistivity and rapid decrease of magnetoresistance. Our discoveries indicate an unusual competition between superconductivity and CDW state in pressurized kagome lattice.

[1] Hefei National Laboratory for Physical Sciences at Microscale and Department of Physics, and CAS Key Laboratory of Strongly-coupled Quantum Matter Physics, University of Science and Technology of China, Hefei, Anhui, China. [2] CAS Center for Excellence in Quantum Information and Quantum Physics, Hefei, Anhui, China. [3] Collaborative Innovation Center of Advanced Microstructures, Nanjing, PR China. ✉email: yingjj@ustc.edu.cn; chenxh@ustc.edu.cn

Superconductivity and the charge density wave (CDW) state are two different cooperative electronic states, and both of them are originated from electron–phonon coupling and Fermi surface instabilities. Superconductivity is often observed in CDW materials, however, the interplay between CDW order and superconductivity is still not well elucidated[1–5]. For example, the maximum $T_c$ in compressed TiSe$_2$ or Cu$_x$TiSe$_2$ is reached when the CDW order is completely suppressed[2,6]; however, the superconductivity in compressed 1$T$-TaS$_2$ and 2$H$-NbSe$_2$ seems to be independent of the CDW order[7,8].

The materials with kagome lattice provide a fertile ground to study the frustrated, novel correlated, and topological electronic states owing to unusual lattice geometry[9–12]. Recently, a new family of quasi two-dimensional kagome metals $A$V$_3$Sb$_5$ ($A$ = K, Rb Cs) has attracted tremendous attention[13]. These materials crystallize in the $P6/mmm$ space group, forming layers of ideal kagome nets of V ions coordinated by Sb. Besides topological properties, $A$V$_3$Sb$_5$ exhibits both CDW[13,14] and superconductivity[15–17]. Ultra-low temperature thermal conductivity measurements on CsV$_3$Sb$_5$ single crystal shows a finite residual linear term, implying an unconventional nodal superconductivity[18]. Intertwining the superconductivity and CDW state by involving nontrivial topology of band structures results in many exotic properties in this type of materials. For example, topological charge order was reported in KV$_3$Sb$_5$[14], signatures of spin-triplet superconductivity were claimed in Nb-K$_{1-x}$V$_3$Sb$_5$ devices[19], and unconventional giant anomalous Hall effect was observed in K$_{1-x}$V$_3$Sb$_5$ and superconducting CsV$_3$Sb$_5$[20,21]. However, the superconducting transition temperature ($T_c$) in this system is relatively low, and its correlation with the CDW and non-trivial topological states are still not known.

High pressure is a clean method to tune the electronic properties without introducing any impurities, and pressure is often used as a control parameter to tune superconductivity and CDW state. Here, we performed high-pressure electrical transport measurements on CsV$_3$Sb$_5$ single crystals with the highest $T_c$ of 2.7 K in $A$V$_3$Sb$_5$ family. Maximum $T_c$ of 8 K is observed at P2 ≈ 2 GPa when CDW is completely suppressed. Strikingly, an unusual suppression on superconductivity is observed between P1 ≈ 0.7 GPa and P2 ≈ 2 GPa. These results indicate an unexpected, exotic competition between CDW and superconductivity in this region, which makes CsV$_3$Sb$_5$ a rare platform to investigate the interplay of multiple electronic orders.

## Results

**High pressure resistivity and magnetic susceptibility measurements.** We performed resistivity and magnetic susceptibility measurements on CsV$_3$Sb$_5$ under pressure to track the evolution of the CDW state and superconductivity in the compressed material. Temperature dependence of resistivity for CsV$_3$Sb$_5$ under various pressures is shown in Fig. 1. As shown in Fig. 1a, an anomaly due to the CDW transition in the resistivity[15] is clearly visible for sample 1 with PCC. The CDW transition temperature $T^*$ gradually decreases with increasing the pressure, and the anomaly becomes much weaker at high pressures. We can determine the CDW transition temperature $T^*$ precisely in the derivative resistivity curves as shown in Fig. 1b. The anomaly clearly shifts to lower temperatures with increasing the pressure, and disappears at the pressure of P2 ≈ 2 GPa. The anomaly of $d\rho_{xx}/dT$ at $T^*$ gradually evolves from a broad peak to a peak-dip feature around P1. Further increasing the pressure above P1, the peak gradually weakens and the dip becomes more pronounced. The $d\rho_{xx}/dT$ anomaly is associated to the change of the band structure and electron scattering rate in the CDW state.

Therefore, such change of the shape of $d\rho_{xx}/dT$ anomaly indicates the possible change of the CDW transition around P1. To track the evolution of $T_c$ with pressure, the low-temperature resistivity of CsV$_3$Sb$_5$ under various pressures is shown in Fig. 1c, d for sample1 with PCC and sample 2 with DAC, respectively. The $T_c$ first increases with increasing pressure as shown in Fig. 1c, however, the superconducting transition becomes much broader with pressure larger than P1 of ~0.7 GPa, reminiscent of an inhomogeneous superconductivity in which multi superconducting phases coexists. In the high-pressure magnetic susceptibility measurements for sample 3 with PCC as shown in Supplementary Fig. 2, the superconducting transition temperature $T_c^{M2}$ determined by the magnetic susceptibility gradually increases with increasing the pressure below P1. With the pressure above P1, the superconducting volume fraction suddenly decreases. Further increasing the pressure, the bulk superconductivity $T_c^{M2}$ emerges below 4 K around 1.1 GPa, but the magnetic susceptibility shows a weak reduction at $T_c^{M1}$ with higher temperature, indicating an inhomogeneous superconductivity, consistent with our resistivity measurements. It is striking that the superconducting transition becomes sharp again around P2. Meanwhile the highest $T_c$ of 8 K is obtained. It should be addressed that the maximum $T_c$ achieved at high pressure is 3 times higher than that (2.7 K) at ambient pressure. With further increasing the pressure, the $T_c$ is monotonically suppressed and completely disappears above 12 GPa as shown in Fig. 1d for sample 2 with DAC. The pressure inhomogeneity or pressure induced disorder may cause the broadening of the superconducting transition. Such extrinsic effects become more pronounced with increasing the pressure. However, in our case, the superconducting transition at P2 is very sharp, which indicates that such extrinsic effect is negligible at least below P2 and the broadening of the superconducting transition between P1 and P2 is intrinsic.

Figure 2a, b shows superconducting transition under different magnetic fields for sample 1 with PCC around P1 and P2 with magnetic field applied along c axis, respectively. Much higher magnetic field is needed to suppress superconductivity around P1 and P2 compared with that at ambient pressure[15,18]. To investigate the evolution of $H_{c2}$ under pressure, we plot $H_{c2}$ (determined by $T_c^{zero}$) as a function of temperature under various pressures, as shown in Fig. 2c, d. The $H_{c2}$ shows strong pressure dependence. $H_{c2}$ dramatically increases with increasing the pressure, and reaches a local maximum value around P1. With further increasing the pressure, the $H_{c2}$ can be rapidly suppressed, as shown in Fig. 2c. When the pressure reaches P2, $H_{c2}$ dramatically increases to the maximum value, then $H_{c2}$ can be gradually suppressed with further increasing the pressure, as shown in Fig. 2d.

**Phase diagram with pressure for CsV$_3$Sb$_5$ single crystal.** Combing the high-pressure electrical and magnetic susceptibility measurements on sample 1 and 3 with PCC and sample 2 with DAC, we can map out the phase diagram of CsV$_3$Sb$_5$ with pressure, as shown in Fig. 3a, b. The CDW is monotonically suppressed with increasing pressure, and the maximum $T_c$ locates at the endpoint of CDW. A subsequent monotonic reduction of $T_c$ is followed at higher pressure. The relatively high $T_c$ (8 K) achieved at high pressure makes it the record in the kagome lattice materials. Such competition between CDW state and superconductivity is usual since the gap opening at CDW state would dramatically reduce the density of states at Fermi surfaces, leading to the suppression of superconductivity within Bardeen-Cooper-Schrieffer (BCS) scenario. It is striking that the superconducting transition becomes much broad with pressure

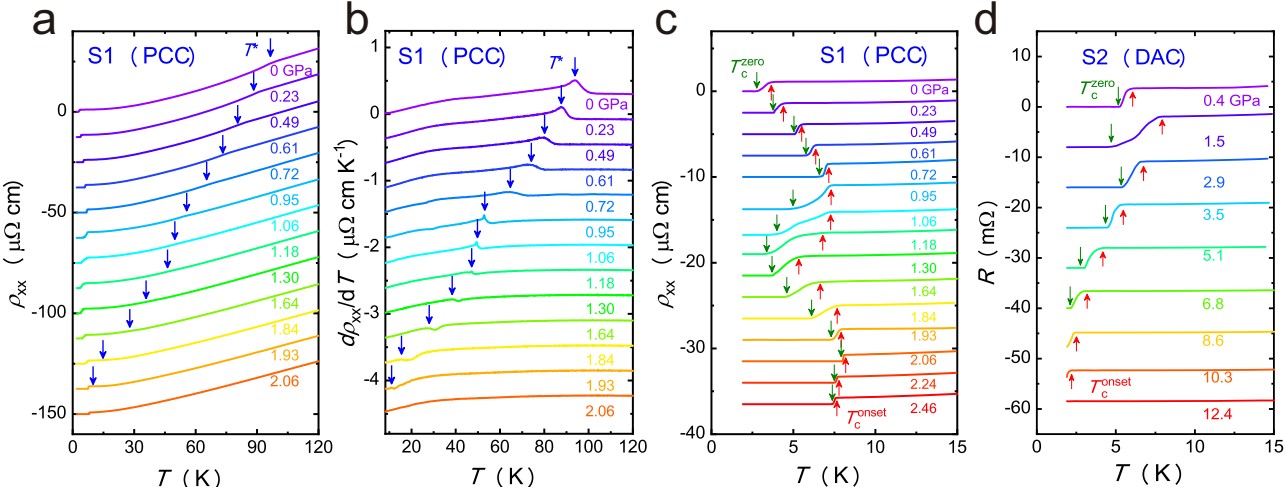

**Fig. 1 Temperature dependence of resistivity in CsV$_3$Sb$_5$ single crystals under high pressure. a** Temperature dependences of resistivity under various pressures for sample 1 measured with PCC. **b** The derivative $d\rho_{xx}/dT$ curves under various pressures. The blue arrows indicate the CDW transition temperature $T^*$. **c**, **d** The evolution of superconducting transition temperatures under pressure for sample 1 with PCC and sample 2 with DAC. The red and green arrows represent the $T_c^{onset}$ and $T_c^{zero}$, respectively. All the curves were shifted vertically for clarity.

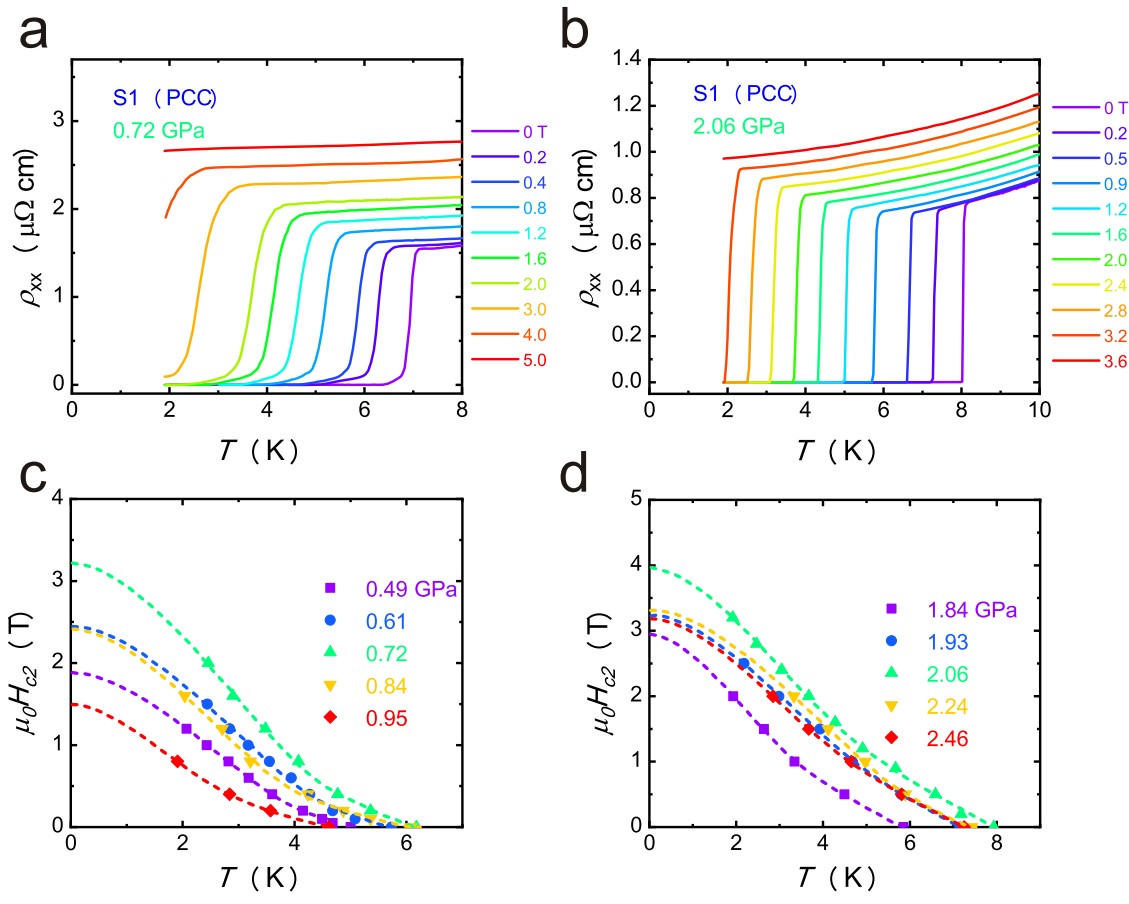

**Fig. 2 Upper critical field for CsV$_3$Sb$_5$ single crystal under various pressures.** Upper critical field ($H_{c2}$) measurements for sample1 around P1 (**a**) and P2 (**b**) with magnetic field applied along $c$ axis. **c**, **d** The upper critical field $\mu_0 H_{c2}$ vs. $T$ under various pressures. The dashed lines are the fitting curves by using the two-band model.

between P1 and P2, and $T_c$ together with $H_{c2}$ are strongly suppressed. The $T^*$ shows a weak anomaly around P1, which suggests the transformation of CDW state. Above P1, the superconductivity shows much stronger competition with CDW order, which will be discussed later. We can estimate the $H_{c2}$ at zero temperature ($H_{c2}(0)$) by using the two-band model[22] of $H_{c2}$ vs $T$ curves as shown in Fig. 2c, d. The pressure dependence of the extracted $H_{c2}(0)$ is shown in Fig. 3c, and two peaks are clearly located at P1 and P2. The magnitude of $H_{c2}(0)$ at P1 and P2 is one order larger than that at ambient pressure.

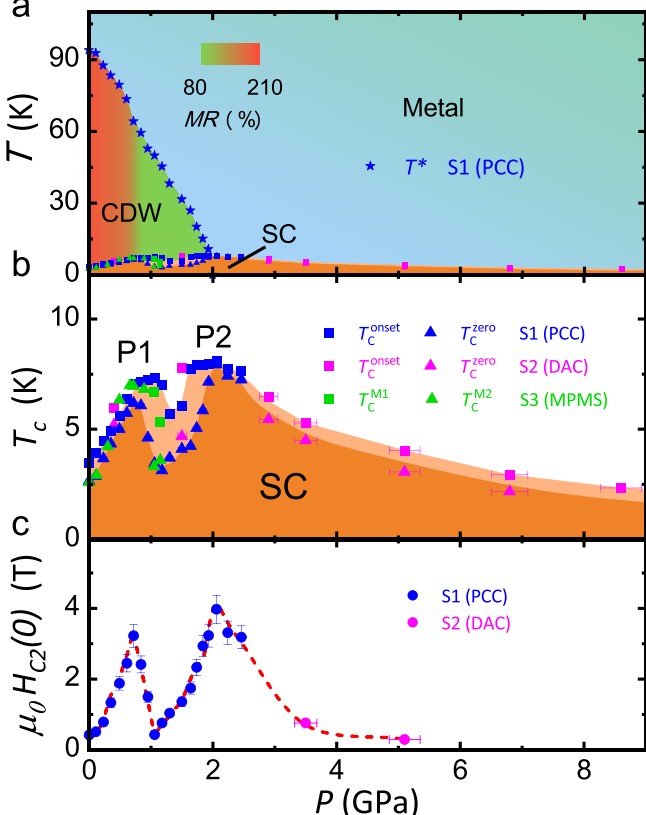

**Fig. 3 Phase diagram with pressure for CsV₃Sb₅ single crystal. a** Phase diagram of $CsV_3Sb_5$ with pressure. CDW transition temperature $T^*$ gradually suppressed with increasing the pressure. $T^*$ shows an anomaly around P1. The color inside the CDW phase represents the magnitude of magnetoresistance measured at 9 T and 10 K. **b** Pressure dependence of superconducting transition temperatures $T_c^{onset}$, $T_c^{zero}$, $T_c^{M1}$ and $T_c^{M2}$ measured on various samples. $T_c$ is strongly suppressed with the pressure between P1 and P2. **c** Pressure dependence of upper critical field at zero temperature, two peaks can be observed at P1 and P2.

**Two transitions at P1 and P2**. The pressure dependence of $T_c^{zero}$ clearly shows that two peaks are located at P1 and P2, and the superconducting transition width $\Delta T_c$ shows sudden enhancement with pressure between P1 and P2, as shown in Fig. 4a. In order to unveil the unusual suppression of superconductivity between P1 and P2, we plot the residual resistivity and residual-resistivity ratio (RRR) as a function of pressure, as shown in Fig. 4b. The residual resistivity suddenly increases around P1 and keeps at a relatively high value with the pressure between P1 and P2. Above P2, the residual resistivity suddenly drops. RRR also exhibits reduction with pressure between P1 and P2. The magnetoresistance (MR) measured under magnetic field of 9 Tesla and at the temperature of 10 K suddenly drops around P1. The MR measured at 3 T shows an anomaly at P1 and sudden drops at P2. In addition, the magnetoresistance of the low-field region at 10 K evolves from "V" shape to "U" shape at P2 as shown in Supplementary Fig. 4. Temperature dependence of MR at high pressure also evolves from "V" shape to "U" shape as shown in Supplementary Fig. 5. These results indicate that the low-field linear MR is an intrinsic property of the CDW phase, since the MR exhibits quadratic temperature-dependence when the CDW is suppressed by both of heating and pressure. However, the room-temperature resistivity gradually decreases with increasing the pressure, and does not show any anomaly at P1 and P2 as shown in Fig. 4c. These results indicate that the transitions at P1 and P2 are related to the CDW transitions rather than the

normal state change at high temperature. The Shubnikov-de Haas (SdH) quantum oscillations (QOs) measurements at 2 K exhibit a rapid damping above P1 as shown in Supplementary Fig. 6, which is possibly associated to the enhancement of scattering rate for the resolved four bands, providing evidence for a transformation of the CDW state at P1.

## Discussion

Although superconductivity was reported in some kagome lattice materials[23,24], the previously reported $T_c$ is quite low. Our high-pressure work demonstrates that the $T_c$ in this V-based kagome material can be relatively high and easily tuned. Further enhancement of $T_c$ should be possible in this type of materials by using the other methods to destroy the CDW state, such as chemical substitution or electrical gating. The maximum $T_c$ locates at the end-point of CDW, which resembles many other CDW materials[2,6], indicating the competition between superconductivity and CDW state. The most interesting discovery in this work is the region between P1 and P2, in which the CDW and superconductivity have much stronger competition, leading to the dramatical suppression of $T_c$ and increment of superconducting transition width ($\Delta T_c$). In addition, the sudden enhancement of residual resistivity and damping of QOs indicate a transition at P1, possibly due to the formation of a new CDW state. One possibility is that the original modulation pattern may change under pressure, and a new commensurate CDW (CCDW) state or an incommensurate CDW (ICCDW) state emerges above P1. Pressure-induced CCDW to ICCDW transition has been observed in the $2H$-$TaSe_2$[25]. Such phase transition will dramatically modify the Fermi surface and alters $T_c$. However, it cannot explain the superconducting transition broadening after phase transition. Another possibility is that a nearly commensurate CDW (NCCDW) state forms above P1, in which CDW domains are separated by domain walls (DW). Similar transition was also observed in $1T$-$TaS_2$ with increasing the temperature[26]. It is proposed a tri-hexagonal (inverse Star of David) modulation pattern shows up in the CCDW state for $AV_3Sb_5$[14]. In the NCCDW state, the charge may transfer to the narrow DWs since CDW domains are partially gapped[27]. The enhanced interaction and scattering at the DW network will lead to the higher resistivity[27]. The superconductivity in the CDW domains may be strongly suppressed due to the reduction of carrier density, however, the superconductivity in the DWs may have higher $T_c$, leading to the inhomogeneous superconductivity. To be reminded, the rather broad superconducting transition in the NCCDW state actually resembles the manifestation of so-called pair-density-wave (PDW) order in high-$T_c$ cuprate superconductors[28]. In $La_{2-x}Ba_x$-$CuO_4$, superconductivity is greatly suppressed around 1/8 doping level due to the formation of a long-ranged stripe order and the final superconducting state at low temperatures is attributed to a PDW order[29–31]. This is very similar to our cases around the region with unusual suppression of $T_c$. Therefore, it is speculated that a similar PDW order might also emerge in the superconducting state between P1 and P2, which needs further evidences from spatial-resolved spectroscopy. Our discoveries may help to clarify the origin of similar behaviors observed in other systems with competing orders. In fact, two-dome $T_c$ behavior were also observed in some superconductors[32–34], which might be a general feature for superconductors with competing density waves. However, we would like to note that two-dome SC behavior in $CsV_3Sb_5$ occurs within the CDW phase, which is different with the other systems where a second SC dome appears when the density wave order is completely suppressed. Quantum critical behavior can emerge at the endpoint of CDW order[35]. In order to search for the quantum criticality in this system, we analysis the low-temperature resistivity, as shown in Supplementary Fig. 8. Above P2, the low-temperature resistivity

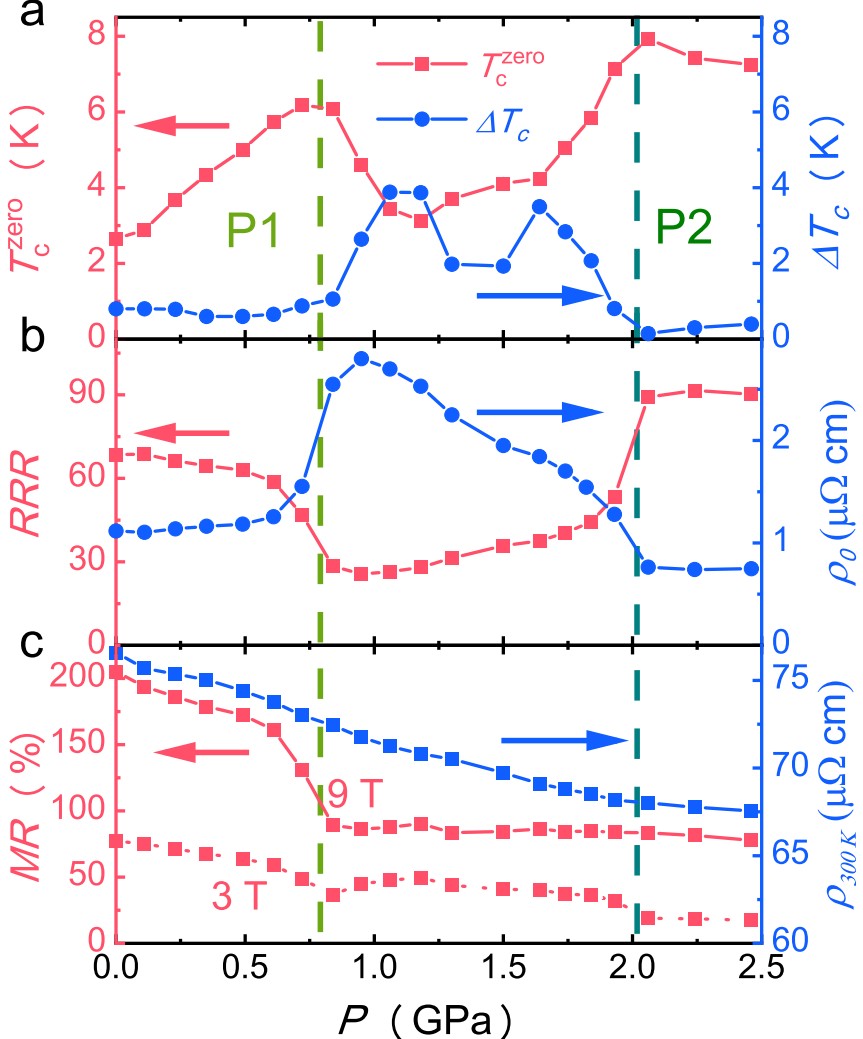

**Fig. 4 Pressure dependence of residual resistivity, residual-resistivity ratio and magnetoresistance for CsV$_3$Sb$_5$ single crystal. a** Pressure dependence of $T_c^{zero}$ and $\Delta T_c$ for sample 1 with PCC. $T_c^{zero}$ exhibits two peaks at P1 and P2. **b** Pressure dependence of residual resistivity and residual resistivity ratio (RRR) for sample 1 with PCC. **c** Pressure dependence of magnetoresistance (9 T and 3 T, 10 K) and room-temperature resistivity.

follows $T^2$ behavior below 35 K, therefore our current results do not support the existence of quantum fluctuations around P2. However, ultralow-temperature experiments are still highly required to clarify possible quantum criticality at lower temperature.

In conclusion, we systematically investigate the high-pressure transport and magnetic susceptibility properties of the newly discovered topological kagome metal CsV$_3$Sb$_5$. When CDW is completely suppressed, the maximum $T_c$ up to 8 K can be reached, and is three times higher than that (2.7 K) at ambient pressure. More interestingly, superconductivity shows an unusual suppression with pressure between P1 and P2. A transformation of CDW state occurs above P1 accompanied with the rapidly damping of QOs, sudden enhancement of residual resistivity and the rapid decrease of magnetoresistance.

## Methods

**Material syntheses**. Single crystals of CsV$_3$Sb$_5$ were synthesized via a self-flux growth method similar to the previous reports[15]. In order to prevent the reaction of Cs with air and water, all the preparation processes were performed in an argon glovebox. After reaction in the furnace, the as-grown CsV$_3$Sb$_5$ single crystals are stable in the air. The excess flux is removed using water and millimeter-sized single crystal can be obtained.

**High-pressure measurements**. Piston cylinder cell (PCC) was used to generate hydrostatic pressure up to 2.4 GPa for high-pressure electrical transport

measurements. Daphne 7373 was used as the pressure transmitting medium in PCC. The pressure values in PCC were determined from the superconducting transition of Sn[36]. Diamond anvil cell (DAC) was used to generate the pressure up to 12 GPa. Diamond anvils with 500 μm culet and c-BN gasket with sample chambers of diameter 200 μm were used. Four Pt wires were adhered to the sample and NaCl was used as the pressure transmitting medium. Pressure was calibrated by using the ruby fluorescence shift at room temperature[37]. Electrical transport measurements were carried out in a Quantum Design physical property measurement system. High-pressure magnetic susceptibility measurements were performed in a miniature PCC using a SQUID magnetometer (MPMS-5 T, Quantum Design).

*Note added*: During the preparation of this manuscript, we noticed a similar high-pressure work[38].

## Data availability

All data supporting the findings of this study are available from the corresponding authors upon reasonable request.

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

## Acknowledgements

This work was supported by the Anhui Initiative in Quantum Information Technologies (Grant no. AHY160000), the National Key Research and Development Program of the Ministry of Science and Technology of China (Grants nos. 2019YFA0704901 and 2017YFA0303001), the Science Challenge Project of China (Grant no. TZ2016004), the Key Research Program of Frontier Sciences, CAS, China (Grant no. QYZDYSSWSLH021), the Strategic Priority Research Program of the Chinese Academy of Sciences (Grant no. XDB25000000), the National Natural Science Foundation of China (Grants nos. 11888101 and 11534010), and the Fundamental Research Funds for the Central Universities (WK3510000011 and WK2030020031).

## Author contributions

X.H.C. and J.J.Y. conceived and designed the experiments. F.H.Y. performed high-pressure electrical transport measurements with the assistance from J.J.Y., D.H.M, W.Z.Z., S.Q.L., X.K.W., and B.L. F.H.Y. synthesized the CsV$_3$Sb$_5$ single crystal. J.J.Y., F.H.Y., and X.H.C. analyzed and interpreted the data. J.J.Y. and X.H.C. wrote the manuscript. All authors discussed the results and commented on the manuscript.

## Competing interests

The authors declare no competing interests.
