## [Peer Review File · Nature Communications]

REVIEWER COMMENTS

Reviewer #1 (Remarks to the Author):

The authors of this manuscript report on the observation of pressure induced changes in the superconductivity of a newly discovered kagome metal, CsV₃Sb₅. The central claim is that pressure initially enhances the superconducting transition temperature while simultaneously suppressing the charge density wave order in this compound; however beyond this initial enhancement, they observe an unconventional suppression of T_c followed by a second enhancement, prior to being suppressed entirely at higher pressures. The result is a double superconducting "dome" in the pressure-driven superconducting phase diagram of this material.

The key observation of the double-dome-type behavior suggests an unconventional superconducting ground state and the authors claim supporting evidence for a Lifshitz transition underpinning this behavior. The observation of weak, filamentary superconductivity at intermediate pressures is used to further support this conjecture. Overall, the topic and central claims of the paper are of sufficient novelty and broad community interest to merit consideration in Nature Communications.

There are, however, a few central claims and issues with the manuscript that preclude recommendation for publication in Nature Communications in its current form. I list these below.

(1) There have been several recent reports depicting a qualitatively different picture of the pressure-driven phase diagram in this class of kagome metals. For instance arXiv:2103.13759 which shows an extended high-pressure dome in CsV₃Sb₅ and arXiv:2102.10959 for a single dome in KV₃Sb₅. A similar double dome is noted in this paper as also being observed in CsV₃Sb₅ in arXiv:2102.09328. Can the authors provide analysis which can preclude extrinsic or disorder-driven effects in their present observation of a two-peak superconducting dome? I think this will be important for the paper and the field.

(2) The conjecture of a Lifshitz transition is not immediately obvious from the quantum oscillation data in the supplementary material. The oscillations are seemingly rapidly damped, which is potentially due to pressure-induced disorder/damping effects. The frequency shifts observed prior to their disappearance are not immediately indicative of a Lifshitz transition to this referee. This claim should either be modified or better justified within the text.

Minor comments:

(3) The paper use some editorial review correcting minor typos/grammar issues throughout.

(4) In Fig. 1, correlating colors between plots in the extended and zoomed-in T ranges would be helpful for the reader. It is a little confusing to trace the CDW and SC transitions in the current presentation at fixed pressure values.

In summary, I believe the manuscript has the potential of presenting a finding of sufficient interest and novelty for Nature Communications, but these points/deficiencies must be satisfactorily addressed first. In particular if points (1) and (2) can be addressed, then the paper would be a valuable addition to a seemingly quickly growing field.

Reviewer #2 (Remarks to the Author):

This is a very interesting paper reporting the pressure phase diagram in the CsV₃Sb₅ superconductor with kagome lattice. The authors find a double-peak structure in the superconducting transition temperature T_c as a function of pressure. The second peak at P2 corresponds to the end point of charge density wave (CDW) order, which can be similarly found in other superconductors with CDW. What is novel here is the observation of the first peak at P1, which is deep inside the CDW phase. To discuss the origin of this unusual peak inside the CDW phase, the authors performed magneto-transport measurements from which they find rapid changes in the Shubnikov-de Haas oscillations and the magnitude of magnetoresistance near P1. I find that these observations are intriguing and the discussion on the possible change in the CDW pattern that affects the superconductivity merits a timely publication in Nature Communications. I recommend that the authors consider the following minor comments before publication.

- 1) The resistivity anomalies at the CDW transition temperature look different between low and high pressure regions. The temperature derivative shown in Fig.1b shows a peak structure below P1, but it shows a dip above 1.3 GPa. Can the author discuss the origin of such a change?
- 2) The authors mention on the filamentary-like superconductivity in the pressure region between P1 and P2 from the width of superconducting transition. However, it would be more informative if the authors show the transition in fields.
- 3) The relationship between superconductivity and the quasiparticle mass is also an important point. Can the authors comment on how the effective mass changes with pressure from the temperature dependence of quantum oscillations?

Reviewer #3 (Remarks to the Author):

The manuscript by Yu et al consolidates existing evidence of the evolution of the CDW and SC transitions in CsV₃Sb₅ with pressure and magnetic field. The present study compares electrical transport measurements on two different pressure experiments, PCC and DAC, and finds qualitatively similar albeit some quantitative differences in the critical parameters for the two.

I have to state upfront that I find little new physics that the current study adds, to warrant publication in Nature Communications: CDW and SC in this and related compounds have already been reported (refs 15 and 21 particularly refer to the Cs compound, while refs 14-21 all are reports on AV₃Sb₅ compounds and their transport properties). I will refer to some more specific points below, but I cannot recommend this study (or even a revised version of it) for publication in Nature Communications.

I have a problem with the existing literature and the evidence of bulk SC in CsV₃Sb₅: in both refs 15 and 21, attempts are made to use thermodynamic measurements to demonstrate that the ~ 2 K transition is bulk SC. However, the jump in specific heat $\Delta C/\gamma T$ (fig. 2f in ref 15) is only $\sim 5/40$, far from a convincing 1.4 value for BCS. If not BCS, there is little to now discussion for the reason for the minute specific heat jump to prove bulk SC. One should be aware of possible secondary phases that can result in the small SC signal: V can show SC in a wide T range (1.8 - 6 K at ambient and applied pressure), CsV (+1.5 K), Sb (2.6 - 3.4 K), Sb_{0.01}-_{0.03}V_{0.99}-_{0.97} (2.6-3.7 K) can all show SC. (Roberts, J. Phys. Chem. Ref. Data, 5, 581 (1976))

While previous reports may demonstrate the CDW transition around 90 K, the present study makes little reference to that evidence. In this paper, the authors show only resistivity data, and the

respective derivatives. The small feature only visible in dr/dT (and not $r(T)$ itself) can be due to small structural distortions, not necessarily accompanied by charge order (CDW). Therefore any discussion of transport data must refer to any existing proof that this is indeed a CDW transition.

To the point of unusual competition between SC and CDW in this compound, I can maybe see a qualitative change in the sharpness of the low T transition in S1 (Fig. 1a). But what the authors call $P1 = 0.7$ GPa and $P2 = 2$ GPa as the critical pressure values where the transition changes from sharp to broad and back, these values are not rigorously justified. How broad is broad, and what is a "sharp" transition? ($\Delta T = ?$) If I looked only at Fig. 1a, I would say the transition is broader for $P = 1.06 - 1.3$ GPa, but sharper otherwise.

if there is indeed a double transition at $P1$ and $P2$ that should be discerned in the phase diagram, the authors must show a more detailed analysis and discuss the physical origin once the two are proven intrinsic to their sample, and not a result of twinning, or secondary phase.

In the present manuscript, the authors report an unconventional two-dome like dependence of superconducting T_c in CsV_3Sb_5 on pressure, concomitant with the suppression of the CDW temperature T^* , with a significant broadening of the superconductive transition in between the domes. The authors attribute this behavior to the presence of a Lifschitz transition caused by the formation of domains close to a near commensurate-commensurate CDW transition.

While these observations certainly add an interesting chapter to the rapidly unfolding CsV_3Sb_5 story, I find the analysis presented in the paper to be insufficient to reach the presented conclusions. In particular, the Lifschitz transition is inferred from the change of frequencies in SdH oscillations with pressure; however, the amplitude of the oscillations is strongly suppressed at the relevant pressure and some of the oscillations could be simply not resolved. Moreover, (see below), no characteristic change in SdH frequencies is demonstrated close to the supposed Lifshitz transition. The broadening of the resistive transition is attributed to the filamentary superconductivity, however, the broadening may also be present due to pressure inhomogeneity and, moreover, no comparison with the Meissner effect is given, which is important for reaching this conclusion unambiguously. Finally, the possibility of NCCDW-CCDW transition is based on a weak anomaly that T^* shows around pressure P1. If present, this transition should exist over a range of temperatures, which is not demonstrated. Thus, all of the main points of the Authors' interpretation require further analysis and argumentation. Additionally, no quantitative fitting/analysis of the SdH oscillations or the magnetoresistance data is given, and the linear extrapolation used (without explanation) for $H_{c2}(T)$ is in clear contrast to the actual data that is noticeably nonlinear.

Therefore, I believe a strongly improved data analysis (see suggestions below) is necessary before the manuscript can be considered for publication in Nature Communications. Below I provide the particular points that the Authors, in my opinion, should address:

(1) The shape of the anomaly of $d\rho/dT$ at T^* (Fig. 1) clearly changes as a function of pressure - from a broad peak to sharp peak to a dip. This behavior should be studied systematically (e.g. by fitting the anomaly shape) and explained in connection to the other results.

(2) The reduction of RRR between P1 and P2 is interpreted as being due to an enhanced scattering rate. However, (1) CsV_3Sb_5 has several bands [PRL 125, 247002 (2020)]; as the respective Fermi energies may depend on pressure without changing the total charge, this may also lead to the observed enhancement of RRR, if the scattering rates for different bands are different. (1a) This is further exacerbated by the author's claim of a Lifshitz transition which implies a strong reduction in

carrier density of one of the bands. Moreover, (2) a mass enhancement near a CDW end point is in principle possible if it is a quantum critical point [see, e.g., J. Phys.: Condens. Matter 13 R723 (2001)] - this will also affect RRR. These scenarios should all be addressed - see a suggestion based on SdH oscillations below. Another option is to measure the Hall resistance and fit $\rho(T)$ and Hall data together with a multiband model.

(3) Apart from RRR and ρ_0 an analysis of the pressure evolution of $\rho(T)$ is needed: does the system show a Fermi liquid-like behavior (at least for low T) $\rho(T) \sim \rho_0 + AT^2$ everywhere in the phase diagram? How does the functional form of $\rho(T)$ change with pressure?

(4) Analysis of the MR data should be performed. The linear MR at low pressures may arise due to linear band crossings [Phys. Rev. B 58, 2788 (1998)], indeed present in CsV₃Sb₅ (Dirac points, see PRL 125, 247002 (2020)) although other explanations [PRL 117,256601 (2016) and refs therein], possibly related to CDW [PNAS June 4, 2019 116 (23) 11201-11206] are also possible. Additionally, the shape change between 1.93 and 2.06 GPa is quite dramatic, while not seen at 9T (Fig. 3c). One possible reason is that the onset of CDW below P2 leads to a reconstruction of the Fermi Surface - evidence for this scenario must come from SdH or Hall effect measurements.

As for the kink around P1 Fig. 3c, to claim a relation between it and CDW a temperature-dependent MR should be presented - there may be something else occurring between 300 K and low-temperature regime.

(5) The claim of a Lifshitz transition is based on the disappearance of some of the peaks in the Fourier spectra of SdH oscillations, Figure S3c. However, no further analysis of SdH oscillations is performed. First, the dependence of the oscillation frequencies on pressure is not reported. In the vicinity of a Lifshitz transition, the corresponding frequencies should be strongly pressure-dependent [for an unrelated example see e.g., Phys. Rev. Lett. 115, 186403 (2015) or npj Quant Mater 4, 2 (2019)]; in contrast to that, no change in frequencies (especially the ones that disappear at 0.75 GPa) is observed between 0 GPa and 0.36 GPa.

Second, the amplitude of the oscillations contains important information about the scattering rates (via the Dingle factor $\delta R \sim e^{-\pi/(\omega_c\tau)}$) and effective masses of the bands (via the Lifshitz-Kosevich temperature dependence). The latter may potentially reveal the mass enhancement near the CDW end point.

In particular, an enhanced scattering rate may simply suppress the oscillations making them barely visible at 0.75 GPa without a Lifshitz transition. Indeed, a strong overall peak height suppression is seen at 0.75 GPa. On the other hand, between 0 GPa and 0.36 GPa, the peak at 30 T seems strongly enhanced, while ρ_0 (Fig. 4b) monotonically grows. These trends should be analyzed (suggestion above)

and discussed.

(6) The broadening of the resistive transition between P1 and P2 is attributed to filamentary-like superconductivity. However, there are other possible explanations of this behavior. First, given the strong $T_c(P)$ dependence any inhomogeneity of pressure in the sample would lead to a broadening of the transition. The authors should provide estimates of such inhomogeneity in their experiment to rule this effect out. Indeed, transition broadening under strain has been reported in Sr2RuO4 and attributed to extrinsic effects (Science 13 Jan 2017 Vol. 355, Issue 6321, eaaf9398). Additionally, the enhanced ρ_0 may suggest an enhanced role of disorder that can affect the transition of an unconventional superconductor.

A direct evidence of filamentary nature would have been a separation between the resistive T_c^{zero} and the onset of Meissner effect (see Fig. 3 in Annu. Rev. Condens. Matter Phys. 2019. 10:25–44, for example) - the authors should provide that to prove their claim.

(7) The determination of $H_{c2}(0)$ by a linear extrapolation is clearly in conflict with the data: for all pressures in Fig. 2c,d one observes a noticeably nonlinear $H_{c2}(T)$ behavior, especially pronounced near 0.72 and 0.84 GPa - the linear fit misses there all but two points. A better motivated description of the data should be used; one can start for example with WHH formula [Phys. Rev. 147, 295 (1966)] or its multiband extensions [A. Gurevich, Phys. Rev. B 82, 184504 (2010), Physica C 456 (2007) 160–169]. The convex shape of $H_{c2}(T)$ suggests that a single-band model would not be enough.

(8) The existence of some kind of (possibly CCDW-NCCDW) transition with pressure is proposed in the paper on the basis of T^* showing a weak anomaly around P1. However, this anomaly is not clearly visible in Fig. 3 - the authors should provide additional analysis close to that point to prove the existence of such anomaly.

Furthermore, the proposed CCDW-NCCDW transition as a function of pressure should be a whole line in the phase diagram of Fig. 3a; the consequence of this would be additional anomalies in $\rho(T)$ dependence (unless the critical pressure is completely independent of temperature) - see e.g., Phys. Rev. Lett. 81, 453 (1998), Scientific Reports volume 6, Article number: 24068 (2016). Are such anomalies observed?

(9) Two more related preprints on superconductivity in CsV₃Sb₅ under pressure have recently appeared: arXiv:2103.12507 and arXiv:2103.13759. The latter may be of particular interest for comparison, as pressure-dependent DFT calculations (and the resulting Fermi surfaces) are reported there.

Minor points:

- Fig. 2 c and d: Is T_c on the horizontal axis a typo -should it be simply T ? I suggest to show $H_{c2}(T)$ vs. T plots if this is not the case.

- An enhanced competition between CDW and superconductivity is suggested to exist between P1 and P2. However, T^* monotonically decreases with P in this region, while T_c is nonmonotonic - for two competing states one would expect T_c to monotonically grow instead. Thus this statement seems to contradict the data.

- The data obtained for 1.5 GPa with DAC Fig. 3b yields a noticeably larger T_C^{onset} than the one measured with PCC. The authors should comment on this discrepancy - this may suggest that the broadening is sample-dependent and hence extrinsic.

- Comparison with $\text{La}_{2-x}\text{Ba}_x\text{CuO}_4$ is not very accurate, since the stripe order is the strongest in region of suppressed T_c , unlike T^* in the current study.

- What are the error bars for points in Figs 2c,d; 3 and 4?

Wording/Typos:

p.4 "...that two peaks locate at P1 and P2..." → "...that two peaks are located at P1 and P2..."

"...transition width of ΔT_c ..." → "...transition width ΔT_c ..."

(multiple places) "...sudden drops..." → "...suddenly drops..."

Reviewer #1 (Remarks to the Author):

The authors of this manuscript report on the observation of pressure induced changes in the superconductivity of a newly discovered kagome metal, CsV₃Sb₅. The central claim is that pressure initially enhances the superconducting transition temperature while simultaneously suppressing the charge density wave order in this compound; however beyond this initial enhancement, they observe an unconventional suppression of T_c followed by a second enhancement, prior to being suppressed entirely at higher pressures. The result is a double superconducting “dome” in the pressure-driven superconducting phase diagram of this material.

The key observation of the double-dome-type behavior suggests an unconventional superconducting ground state and the authors claim supporting evidence for a Lifshitz transition underpinning this behavior. The observation of weak, filamentary superconductivity at intermediate pressures is used to further support this conjecture. Overall, the topic and central claims of the paper are of sufficient novelty and broad community interest to merit consideration in Nature Communications.

There are, however, a few central claims and issues with the manuscript that preclude recommendation for publication in Nature Communications in its current form. I list these below.

(1) There have been several recent reports depicting a qualitatively different picture of the pressure-driven phase diagram in this class of kagome metals. For instance arXiv:2103.13759 which shows an extended high-pressure dome in CsV₃Sb₅ and arXiv:2102.10959 for a single dome in KV₃Sb₅. A similar double dome is noted in this paper as also being observed in CsV₃Sb₅ in arXiv:2102.09328. Can the authors provide analysis which can preclude extrinsic or disorder-driven effects in their present observation of a two-peak superconducting dome? I think this will be important for the paper and the field.

Reply: We thank the reviewer for pointing out this issue. In arXiv: 2103.13759, they only used the DAC to generate the pressure. The pressure inhomogeneity is relatively large and pressure cannot be precisely controlled with DAC. For this reason, they do not observe the double dome behavior. While for KV₃Sb₅, it exhibits both lower T* and T_c comparing with CsV₃Sb₅ at ambient pressure. Therefore, the high-pressure phase diagram may be different due to the sample difference.

Pressure inhomogeneity or pressure induced disorder in the sample may make the superconducting transition broad. However, the two-peak superconducting dome reported in our paper cannot origin from these extrinsic effects for the following reasons. These extrinsic effects become more prominent with increasing the pressure. Actually, the pressure inhomogeneity in PCC is rather small. In our case, with the pressure at P2 with PCC, the superconducting transition is very sharp (~0.2K). Therefore, such extrinsic effect is negligible in our experiments and the broadening of the superconducting transition between P1 and P2 could not origin from these extrinsic origins. We added the related discussion in the revised manuscript.

(2) The conjecture of a Lifshitz transition is not immediately obvious from the quantum oscillation data in the supplementary material. The oscillations are seemingly rapidly damped, which is potentially due to pressure-induced disorder/damping effects. The frequency shifts observed prior to

their disappearance are not immediately indicative of a Lifshitz transition to this referee. This claim should either be modified or better justified within the text.

Reply: We thank the reviewer for pointing out this problem. We agree with the reviewer that the rapidly damping above P1 cannot support the conclusion of Lifshitz transition. Instead, it can be related to the enhanced scattering rate above P1, consistent with the sudden increment of the residual resistivity. We modified this claim in the revised manuscript. However, the sudden change of residual resistivity, MR behavior and quantum oscillation support the conclusion of a new CDW state above P1.

Minor comments:

(3) The paper uses some editorial review correcting minor typos/grammar issues throughout.

Reply: We corrected typos/grammar issues in the revised manuscript.

(4) In Fig. 1, correlating colors between plots in the extended and zoomed-in T ranges would be helpful for the reader. It is a little confusing to trace the CDW and SC transitions in the current presentation at fixed pressure values.

Reply: We use the same colors between plots in the extended and zoomed-in T ranges in Fig.1 as the reviewer suggested.

In summary, I believe the manuscript has the potential of presenting a finding of sufficient interest and novelty for Nature Communications, but these points/deficiencies must be satisfactorily addressed first. In particular if points (1) and (2) can be addressed, then the paper would be a valuable addition to a seemingly quickly growing field.

Reviewer #2 (Remarks to the Author):

This is a very interesting paper reporting the pressure phase diagram in the CsV₃Sb₅ superconductor with kagome lattice. The authors find a double-peak structure in the superconducting transition temperature T_c as a function of pressure. The second peak at P2 corresponds to the end point of charge density wave (CDW) order, which can be similarly found in other superconductors with CDW. What is novel here is the observation of the first peak at P1, which is deep inside the CDW phase. To discuss the origin of this unusual peak inside the CDW phase, the authors performed magneto-transport measurements from which they find rapid changes in the Shubnikov-de Haas oscillations and the magnitude of magnetoresistance near P1. I find that these observations are intriguing and the discussion on the possible change in the CDW pattern that affects the superconductivity merits a timely publication in Nature Communications. I recommend that the authors consider the following minor comments before publication.

1) The resistivity anomalies at the CDW transition temperature look different between low and high

pressure regions. The temperature derivative shown in Fig.1b shows a peak structure below P1, but it shows a dip above 1.3 GPa. Can the author discuss the origin of such a change?
 Reply: We thank the referee for pointing out this issue. The derivative curve shows a peak-dip behavior (for example, see 0.95GPa) above P1. Further increasing the pressure, the peak becomes much weaker (but still exist) and the dip becomes more pronounced. The resistivity anomaly is associated to the change for the band structure and electron scattering rate in the CDW state. The change of the shape of $d\rho_{xx}/dT$ anomaly indicates the possibly change of the CDW state. Such behavior is consistent with sudden change of MR and residual resistivity above P1. We added the related discussion in the revised manuscript.

2) The authors mention on the filamentary-like superconductivity in the pressure region between P1 and P2 from the width of superconducting transition. However, it would be more informative if the authors show the transition in fields.

Reply: We have added a figure (Supplementary Fig.3, we also show below for convenience) in the supplementary information to show the superconducting transition in different fields with pressure between P1 and P2. We also performed the high-pressure magnetic susceptibility measurements (Supplementary Fig.2) to support our conclusions. With the pressure around P1, the superconducting volume fraction suddenly decreases. Further increasing the pressure, the bulk superconductivity T_c^{M2} emerges below 4 K around 1.1 GPa, but the magnetic susceptibility shows a weak reduction at T_c^{M1} with higher temperature, indicating a filamentary superconductivity at higher temperature, consistent with our resistivity measurements.

3) The relationship between superconductivity and the quasiparticle mass is also an important point. Can the authors comment on how the effective mass changes with pressure from the temperature dependence of quantum oscillations?

Reply: We extract the effective mass with pressure at 0.36 GPa as shown in Supplementary Fig.7. The extracted effective mass for F1 and F2 orbitals does not change much with pressure. We added related descriptions in the revised manuscript. However, we should point out that it is a multiband system. The evolution of the effective mass for the other Fermi pockets, especially the large Fermi pockets around Γ and K points cannot be determined in our experiments.

Reviewer #3 (Remarks to the Author):

The manuscript by Yu et al consolidates existing evidence of the evolution of the CDW and SC transitions in CsV₃Sb₅ with pressure and magnetic field. The present study compares electrical transport measurements on two different pressure experiments, PCC and DAC, and finds qualitatively similar albeit some quantitative differences in the critical parameters for the two.

I have to state upfront that I find little new physics that the current study adds, to warrant publication in Nature Communications: CDW and SC in this and related compounds have already been reported (refs 15 and 21 particularly refer to the Cs compound, while refs 14-21 all are reports on AV₃Sb₅ compounds and their transport properties). I will refer to some more specific points below, but I cannot recommend this study (or even a revised version of it) for publication in Nature Communications.

Reply: We thank the reviewer's careful review. However, we cannot agree with the reviewer's comments that our paper adds little new physics to the current study. Although SC and CDW is already reported in CsV₃Sb₅, the interplay of SC and CDW is still unknown before our work. Usually, the suppression of CDW order by doping or pressure always enhances superconductivity and leads to a dome-like behavior for superconductivity in most of CDW materials with superconductivity. To our surprise, we find an unusual competition of CDW and SC under pressure. The T_c shows intriguing two-peak behavior which is never observed in the other CDW materials. Such behavior is possibly due to the formation of a new CDW order at high pressure, which have stronger competition with SC. Our discoveries indicate the unconventional interplay of CDW and SC in CsV₃Sb₅, which would stimulate broad interests on the study of the correlation between CDW and SC in many unconventional superconductors, such as cuprate superconductors.

I have a problem with the existing literature and the evidence of bulk SC in CsV₃Sb₅: in both refs 15 and 21, attempts are made to use thermodynamic measurements to demonstrate that the ~ 2 K transition is bulk SC. However, the jump in specific heat $\Delta C/\gamma T$ (fig. 2f in ref 15) is only $\sim 5/40$, far from a convincing 1.4 value for BCS. If not BCS, there is little to now discussion for the reason for the minute specific heat jump to prove bulk SC. One should be aware of possible secondary phases that can result in the small SC signal: V can show SC in a wide T range (1.8 - 6 K at ambient and applied pressure), CsV (+1.5 K), Sb (2.6 - 3.4 K), Sb_{0.01-0.03}V_{0.99-0.97} (2.6-3.7 K) can all show SC. (Roberts, J. Phys. Chem. Ref. Data, 5, 581 (1976))

Reply: Although the mechanism of superconductivity in this material is still not well understood, a bulk superconducting state in CsV₃Sb₅ is already confirmed by different techniques, including magnetic susceptibility, heat capacity, thermal conductivity, STM and so on. Actually, we used the

same batch of samples with arxiv: 2102.10987, arxiv:2103.04760 and arxiv: 2103.11796. In arxiv: 2103.11796, the heat capacity can be well fitted by using the two s-wave model and the bulk superconductivity is confirmed. In arxiv:2103.04760, superconducting gap is observed by STM, directly proves that the SC is from CsV_3Sb_5 rather than the secondary phases. In addition, no impurity phases can be detected from the x-ray diffraction measurements (arxiv: 2102.10987). Our high-pressure susceptibility measurements also confirm the bulk superconductivity in our sample and exclude the possibility of the secondary phases.

While previous reports may demonstrate the CDW transition around 90 K, the present study makes little reference to that evidence. In this paper, the authors show only resistivity data, and the respective derivatives. The small feature only visible in dr/dT (and not $r(T)$ itself) can be due to small structural distortions, not necessarily accompanied by charge order (CDW). Therefore any discussion of transport data must refer to any existing proof that this is indeed a CDW transition. Reply: The anomaly of the resistivity is already confirmed to be associated with the CDW temperature at ambient pressure. We added the related references in the revised manuscript. The CDW transition is already confirmed by the STM and x-ray scattering measurements (arxiv: 2103.04760 and arxiv: 2103.09769). By increasing the pressure, T^* determined by the anomaly of $d\rho_{xx}/dT$ smoothly changes with the pressure. Thus, it is a natural inference that such an anomaly should be related to the CDW transition rather than the other small structural distortions. In addition, our cooperators also confirm the CDW transition at 1.8 GPa from the high-pressure NMR measurements which will be published elsewhere. For the above reasons, T^* can be associated to the anomaly of $d\rho_{xx}/dT$ curves.

To the point of unusual competition between SC and CDW in this compound, I can maybe see a qualitative change in the sharpness of the low T transition in S1 (Fig. 1a). But what the authors call P1 = 0.7 GPa and P2 = 2 GPa as the critical pressure values where the transition changes from sharp to broad and back, these values are not rigorously justified. How broad is broad, and what is a "sharp" transition? ($\Delta T = ?$) If I looked only at Fig. 1a, I would say the transition is broader for $P = 1.06 - 1.3$ GPa, but sharper otherwise.

Reply: The superconducting transition width ΔT_c is shown in Fig.4a. We can clearly see a larger ΔT_c between P1 and P2. Our high-pressure susceptibility measurements indicate filamentary superconductivity in this pressure range. Besides large ΔT_c in this region, MR and the residual resistivity also show sudden changes at P1 and P2. In addition, the SdH QOs damp rapidly above P1. For all the above reasons, P1 and P2 are the critical pressure values.

if there is indeed a double transition at P1 and P2 that should be discerned in the phase diagram, the authors must show a more detailed analysis and discuss the physical origin once the two are proven intrinsic to their sample, and not a result of twinning, or secondary phase.

Reply: Our single crystals are pure and the superconductivity could not be related to the secondary phase as already testified by the other experimental methods in the other papers (arxiv: 2102.10987, arxiv:2103.04760 and arxiv: 2103.11796.). . The pressure inhomogeneity becomes more pronounced with increasing the pressure. In our experiment, the pressure inhomogeneity of the PCC cell is small as evidenced by the sharp SC transition at P2(~ 0.2 K). Thus, the broadening of the superconducting

transition and the double transition at P1 and P2 could not be due to extrinsic effects. The origin of the T_c peak at P1 can be attributed to the emergence of a new CDW state which has much stronger competition with SC. We propose the possible CCDW to NCCDW transition at P1. We added the related discussion in the revised manuscript.

Reviewer #4 (Remarks to the Authors):

In the present manuscript, the authors report an unconventional two-dome like dependence of superconducting T_c in CsV3Sb5 on pressure, concomitant with the suppression of the CDW temperature T^* , with a significant broadening of the superconductive transition in between the domes. The authors attribute this behavior to the presence of a Lifshitz transition caused by the formation of domains close to a near commensurate-commensurate CDW transition.

While these observations certainly add an interesting chapter to the rapidly unfolding CsV3Sb5 story, I find the analysis presented in the paper to be insufficient to reach the presented conclusions. In particular, the Lifshitz transition is inferred from the change of frequencies in SdH oscillations with pressure; however, the amplitude of the oscillations is strongly suppressed at the relevant pressure and some of the oscillations could be simply not resolved. Moreover, (see below), no characteristic change in SdH frequencies is demonstrated close to the supposed Lifshitz transition. The broadening of the resistive transition is attributed to the filamentary superconductivity, however, the broadening may also be present due to pressure inhomogeneity and, moreover, no comparison with the Meissner effect is given, which is important for reaching this conclusion unambiguously. Finally, the possibility of NCCDW-CCDW transition is based on a weak anomaly that T^* shows around pressure P1. If present, this transition should exist over a range of temperatures, which is not demonstrated. Thus, all of the main points of the Authors' interpretation require further analysis and argumentation. Additionally, no quantitative fitting/analysis of the SdH oscillations or the magnetoresistance data is given, and the linear extrapolation used (without explanation) for $H_{c2}(T)$ is in clear contrast to the actual data that is noticeably nonlinear.

Therefore, I believe a strongly improved data analysis (see suggestions below) is necessary before the manuscript can be considered for publication in Nature Communications. Below I provide the particular points that the Authors, in my opinion, should address:

(1) The shape of the anomaly of $d\rho/dT$ at T^* (Fig. 1) clearly changes as a function of pressure - from a broad peak to sharp peak to a dip. This behavior should be studied systematically (e.g. by fitting the anomaly shape) and explained in connection to the other results.

Reply: We thank the reviewer's great suggestion. We added the related discussion in the revised manuscript. The shape of the anomaly of $d\rho_{xx}/dT$ at T^* gradually evolves from the broad peak to peak-dip behavior around P1. Further increasing the pressure, the peak becomes much weaker and the dip becomes much pronounced. The resistivity is strongly connected with band structure and electron scattering. The sudden enhancement of residual resistivity and rapid damping of QOs indicate the enhancement of the scattering rate (possible for some bands), which would naturally lead to the change of the $d\rho_{xx}/dT$ behavior. The change of the shape of $d\rho_{xx}/dT$ anomaly at T^* can be also related to the evolution to a new CDW state, possibly from CCDW to NCCDW.

(2) The reduction of RRR between P1 and P2 is interpreted as being due to an enhanced scattering rate. However, (1) CsV₃Sb₅ has several bands [PRL 125, 247002 (2020)]; as the respective Fermi energies may depend on pressure without changing the total charge, this may also lead to the observed enhancement of RRR, if the scattering rates for different bands are different. (1a) This is further exacerbated by the author's claim of a Lifshitz transition which implies a strong reduction in carrier density of one of the bands. Moreover, (2) a mass enhancement near a CDW end point is in principle possible if it is a quantum critical point [see, e.g., J. Phys.: Condens. Matter 13 R723 (2001)] - this will also affect RRR. These scenarios should all be addressed - see a suggestion based on SdH oscillations below. Another option is to measure the Hall resistance and fit $\rho(T)$ and Hall data together with a multiband model.

Reply: We agree with the referee that the RRR may change depends on the respective Fermi energies of different bands. Actually, the band structure of CsV₃Sb₅ is rather complicated, there are many bands at the Fermi energy. The dependence of each band under pressure is very important but out of the scope of our current work. Actually, the four frequencies resolved from the SdH oscillations only contribute less than 1% of the whole area of Brillouin zone. The large electron Fermi pockets around Γ and K points cannot be resolved in the SdH oscillations. For these reasons, we can not specify the explicit origins of RRR change under pressure. From the QO measurements, the rapidly damping of the QOs above P1 indicates the enhanced scattering rate for the four resolved frequencies. But we cannot figure out the change of the other bands above P1 from the current measurements.

The quantum critical point near the CDW end point is an interesting topic, we checked our data and analyses the $\rho(T)$ at low temperature to check the possibility of quantum critical point in the revised manuscript. Since $\rho(T)$ follows T^2 behavior and ρ_0 is small at P2 as shown in Supplementary Fig.8. Thus, our data does not support the quantum criticality at the CDW end point. Since we cannot detect any QOs around P2, thus the effect mass could not be obtained from the QOs. The square of effective mass is proportional to the parameter A (which can be obtained from the $\rho(T) \sim \rho_0 + AT^2$ fitting). We can clearly see the slope of the ρ_{xx} vs. T^2 curves nearly does not change around P2 shown in Supplementary Fig.8. Therefore, our data does not support for the mass enhancement around P2. However, we should also point out that ultralow-temperature measurements are still needed to check the existence of quantum criticality in the lower temperature region. We added the related discussion in the revised manuscript.

(3) Apart from RRR and ρ_0 an analysis of the pressure evolution of $\rho(T)$ is needed: does the system show a Fermi liquid-like behavior (at least for low T) $\rho(T) \sim \rho_0 + AT^2$ everywhere in the phase diagram? How does the functional form of $\rho(T)$ change with pressure?

Reply: We thank referee's great suggestion. We analyses the $\rho(T)$ at low temperature as shown in Supplementary Fig.8 (We also show the figure below for convenience). Below P1, $\rho(T)$ does not follow the T^2 well, however, with pressure at P2, $\rho(T)$ follows T^2 behavior indicating the Fermi liquid nature. In addition, the residual resistivity is small at P2, therefore, our data does not support the quantum criticality at the CDW end point. However, we should also point out that ultralow-temperature measurements are still needed to check the existence of quantum criticality at lower temperature in the future studies. The related description is added in the revised manuscript.

(4) Analysis of the MR data should be performed. The linear MR at low pressures may arise due to linear band crossings [Phys. Rev. B 58, 2788 (1998)], indeed present in CsV3Sb5 (Dirac points, see PRL 125, 247002 (2020)) although other explanations [PRL 117,256601 (2016) and refs therein], possibly related to CDW [PNAS June 4, 2019 116 (23) 11201-11206] are also possible. Additionally, the shape change between 1:93 and 2:06 GPa is quite dramatic, while not seen at 9T (Fig. 3c). One possible reason is that the onset of CDW below P2 leads to a reconstruction of the Fermi Surface - evidence for this scenario must come from SdH or Hall effect measurements. As for the kink around P1 Fig. 3c, to claim a relation between it and CDW a temperature-dependent MR should be presented - there may be something else occurring between 300 K and low-temperature regime.

Reply: We thank the reviewer's great suggestions. The electronic structure and electron scattering may be quite different in the different pressure regions, which lead to the different MR behavior and RRR. We add the related discussion of the MR data in the revised manuscript. The linear MR at the low field region below P2 can possibly arise from the linear band crossings or/and CDW state. We also added a figure (Supplementary Fig.5, also shown below for convenience) to show the temperature-dependent MR at 1.02 GPa for sample 4 with PCC. The T^* is suppressed to 49K. We can clearly see that the shape of MR sudden changes around T^* , similar with shape change between 1.93 and 2.06 GPa shown in Supplementary Fig.4. These results confirm the low-field linear MR is related to the CDW state. It is possible that a reconstruction of the Fermi Surface may occur in the CDW state. In principle, SdH or Hall effect measurements may give some indication. However, in our case, SdH can be only observed at low temperature and pressure, we cannot extract any QOs across the CDW transition. While for Hall measurements, we can see the carrier density slightly changes around T^* (arxiv: 2102.10987), but cannot prove the Fermi surface reconstruction even at ambient pressure. The Hall resistivity at low temperature becomes very complicated since it is a multiband system and anomalous Hall effect becomes more pronounced. It is still a challenge to understand its Hall effect right now. For these reasons, we cannot give any conclusion of Fermi surface reconstruction at T^* from the current measurements.

(5) The claim of a Lifshitz transition is based on the disappearance of some of the peaks in the Fourier spectra of SdH oscillations, Figure S3c. However, no further analysis of SdH oscillations is performed. First, the dependence of the oscillation frequencies on pressure is not reported. In the vicinity of a Lifshitz transition, the corresponding frequencies should be strongly pressure-dependent [for an unrelated example see e.g., Phys. Rev. Lett. 115, 186403 (2015) or npj Quant Mater 4, 2 (2019)]; in contrast to that, no change in frequencies (especially the ones that disappear at 0.75 GPa) is observed between 0 GPa and 0.36 GPa. Second, the amplitude of the oscillations contains important information about the scattering rates (via the Dingle factor $\delta R \sim e^{-\pi/(\omega c \tau)}$) and effective masses of the bands (via the Lifshitz-Kosevich temperature dependence). The latter may potentially reveal the mass enhancement near the CDW end point. In particular, an enhanced scattering rate may simply suppress the oscillations making them barely visible at 0.75 GPa without a Lifshitz transition. Indeed, a

strong overall peak height suppression is seen at 0.75 GPa. On the other hand, between 0 GPa and 0.36 GPa, the peak at 30 T seems strongly enhanced, while ρ_0 (Fig. 4b) monotonically grows. These trends should be analyzed (suggestion above) and discussed.

Reply: We thank reviewer's great suggestions. We agree with the reviewer that the current data cannot support the Lifshitz transition. Therefore, we modified our description of the Lifshitz transition in the revised manuscript. As the reviewer point out that the enhanced scattering rate can also suppress the oscillations without a Lifshitz transition, the rapid damping of the QOs can be attributed to the enhanced scattering rate for the 4 orbitals observed in our QOs which is consistent with the sudden enhancement of residual resistivity above P1. The QOs damping rapidly above P1. Therefore, we cannot extract the information from QOs near the CDW end point.

We added more discussions about the SdH oscillations in the revised manuscript. We analyzed the SdH oscillations at 0.36GPa at different temperatures to extract the effect mass as shown in Supplementary Fig.7. We found the effect mass does not change much comparing with that at ambient pressure. We also analyzed the enhancement of amplitude for the 26 and 72T orbitals at 0.36 GPa in the revised manuscript. The amplitude of the QOs depends on thermal damping factor R_T and Dingle damping factor R_D . Since the effect mass does not change much at 0.36 GPa, the enhancement of amplitude can be attributed to the reduction of the scattering rate for the two bands.

(6) The broadening of the resistive transition between P1 and P2 is attributed to filamentary-like superconductivity. However, there are other possible explanations of this behavior. First, given the strong $T_c(P)$ dependence any inhomogeneity of pressure in the sample would lead to a broadening of

the transition. The authors should provide estimates of such inhomogeneity in their experiment to rule this effect out. Indeed, transition broadening under strain has been reported in Sr2RuO4 and attributed to extrinsic effects (Science 13 Jan 2017 Vol. 355, Issue 6321, eaaf9398). Additionally, the enhanced ρ_0 may suggest an enhanced role of disorder that can affect the transition of an unconventional superconductor. A direct evidence of filamentary nature would have been a separation between the resistive T_{zeroc} and the onset of Meissner effect (see Fig. 3 in Annu. Rev. Condens. Matter Phys. 2019. 10:25{44, for example) - the authors should provide that to prove their claim.

Reply: The pressure inhomogeneity is a common problem in the high-pressure experiment. It is usually negligible in the PCC by using the liquid pressure medium. The pressure inhomogeneity increases with increasing the pressure, especially when the liquid pressure medium become solidified under pressure. In our case, the superconducting transition is very sharp (~ 0.2 K) at P2 with PCC, which indicates the pressure inhomogeneity is negligible in our experiments. If we assume that the superconducting transition width totally comes from the pressure inhomogeneity (actually it is impossible in reality, but we can use this to give an estimation of upper limit of pressure inhomogeneity in PCC), the upper limit of pressure inhomogeneity can be estimated to be ~ 0.1 GPa around 2 GPa. Thus, the broadening of the superconducting transition between P1 and P2 cannot origin from the small pressure inhomogeneity. We added the related description in the revised manuscript.

We performed the high-pressure magnetic susceptibility measurements as shown in Supplementary Fig.2 (We also show the figure below for convenience). The superconducting volume fraction suddenly decreases above P1. With pressure around 1.1 GPa, two transitions can be observed. The bulk superconducting temperature T_{c1}^{M2} appears below 4 K, however, the magnetic susceptibility also shows a weak reduction at T_{c2}^{M1} which is much higher than T_{c1}^{M2} , representing the filamentary superconductivity, consistent with the resistivity measurements. The related description is added in the revised manuscript.

(7) The determination of $H_{c2}(0)$ by a linear extrapolation is clearly in conflict with the data: for all pressures in Fig. 2c,d one observes a noticeably nonlinear $H_{c2}(T)$ behavior, especially pronounced near 0.72 and 0.84 GPa - the linear fit misses there all but two points. A better motivated description of the data should be used; one can start for example with WHH formula [Phys. Rev. 147, 295 (1966)]

or its multiband extensions [A. Gurevich, Phys. Rev. B 82, 184504 (2010), Physica C 456 (2007) 160-169]. The convex shape of $H_{c2}(T)$ suggests that a single-band model would not be enough.

Reply: We thank the reviewer for pointing out this problem. We totally agree with the reviewer that the convex shape of $H_{c2}(T)$ suggests that a single-band model cannot describe the $H_{c2}(T)$ behavior. Therefore, we used two-band model to fit the $H_{c2}(T)$ in the revised manuscript. The nonlinear $H_{c2}(T)$ behavior can be well fitted by using the two-band model. We can obtain $H_{c2}(0)$ from the fitting of two-band model in the revised manuscript.

(8) The existence of some kind of (possibly CCDW-NCCDW) transition with pressure is proposed in the paper on the basis of T^* showing a weak anomaly around P1. However, this anomaly is not clearly visible in Fig. 3 - the authors should provide additional analysis close to that point to prove the existence of such anomaly. Furthermore, the proposed CCDW-NCCDW transition as a function of pressure should be a whole line in the phase diagram of Fig. 3a; the consequence of this would be additional anomalies in $\rho(T)$ dependence (unless the critical pressure is completely independent of temperature) - see e.g., Phys. Rev. Lett. 81, 453 (1998), Scientific Reports volume 6, Article number: 24068 (2016). Are such anomalies observed?

Reply: We added more discussion of the anomalies around P1. The RRR and MR shows a sudden change at P1. In addition, the SdH oscillations damping rapidly above $\sim P1$. All these information indicate a transition around P1. From the current measurement we can hardly determine any anomalies from $\rho(T)$ except the one around T^* as shown in Fig.1. The domain walls may gradually emerge above P1, but it is not sensitive to the temperature in our case. Another possibility is that such transition may be rather broad, we cannot detect it in the resistivity measurement. In order to solve this problem, detailed high-pressure low-temperature x-ray diffraction measurements are highly required in the future experiment but not in our current work. For example, in compressed TiSe_2 , x-ray diffraction measurements give a complex phase diagram (*Nature Physics* **10**, 421-425 (2014)) including the ICCDW and CCDW in the phase diagram. However, for the resistivity measurements, only one anomaly can be detected from the resistivity curves (*Physical Review Letters* **103**, 236401 (2009).).

(9) Two more related preprints on superconductivity in CsV_3Sb_5 under pressure have recently appeared: arXiv:2103.12507 and arXiv:2103.13759. The latter may be of particular interest for comparison, as pressure-dependent DFT calculations (and the resulting Fermi surfaces) are reported there.

Reply: We thank the reviewer for providing these papers. In these papers, they focused on a new superconducting phase which emerges at much higher pressure (above $\sim 15\text{GPa}$). In our case, the most intriguing discovery was found with pressure below 2GPa. In addition, we are focusing on the unconventional competition of CDW and superconductivity in our paper. In the second superconducting phase reported in arXiv:2103.12507 and arXiv:2103.13759, the CDW order is absent and not taken into consideration in their DFT calculations.

Minor points:

- Fig. 2 c and d: Is T_c on the horizontal axis a typo -should it be simply T ? I suggest to show $Hc_2(T)$ vs. T plots if this is not the case.

Reply: We thank the referee for pointing out this problem. We modified it in the revised manuscript.

- An enhanced competition between CDW and superconductivity is suggested to exist between P1 and P2. However, T^* monotonically decreases with P in this region, while T_c is nonmonotonic - for two competing states one would expect T_c to monotonically grow instead. Thus this statement seems to contradict the data.

Reply: The enhanced competition between CDW and superconductivity is referred to the region above P1 comparing with that below P1. The new CDW state appears above P1 which has stronger competition with superconductivity. We modified the description in the revised manuscript.

- The data obtained for 1.5 GPa with DAC Fig. 3b yields a noticeably larger T_c^{onset} than the one measured with PCC. The authors should comment on this discrepancy - this may suggest that the broadening is sample-dependent and hence extrinsic.

Reply: The pressure in DAC is determined at room temperature. The pressure in our DAC cell has ~5% variation at low temperature. In addition, the pressure inhomogeneity in DAC is larger than that in PCC since we used solid pressure medium, which makes the superconducting transition slightly more broad comparing with that in PCC. However, the broadening of T_c within the P1 and P2 region is also obvious with DAC. Since the superconducting curve above P2 become sharp again which cannot be attributed to the pressure inhomogeneity effect. For these reasons, the observed broadening is intrinsic.

- Comparison with $\text{La}_{2-x}\text{Ba}_x\text{CuO}_4$ is not very accurate, since the stripe order is the strongest in region of suppressed T_c , unlike T^* in the current study.

Reply: We thank the reviewer for pointing out this issue. However, we should point out that the system may evolve to a new CDW state above P1, in which the new CDW state has strong competition with superconductivity, similar to the case in $\text{La}_{2-x}\text{Ba}_x\text{CuO}_4$.

- What are the error bars for points in Figs 2c,d; 3 and 4?

Reply: We added error bars in the revised manuscript.

Wording/Typos:

p.4 "...that two peaks locate at P1 and P2..." → "...that two peaks are located at P1 and P2..."
 "...transition width of ΔT_c ..." → "...transition width ΔT_c ..." (multiple places) "...sudden drops..." → "...suddenly drops..."

Reply: We thank the reviewer for pointing out these typos. We corrected the typos in the revised manuscript.

REVIEWER COMMENTS

Reviewer #1 (Remarks to the Author):

The authors addressed the initial concerns I had about the manuscript. The revised discussion and presentation are now more in line with the data presented. I now support publication in Nature Communications.

Reviewer #2 (Remarks to the Author):

In the revised manuscript the authors address my previous comments in a reasonable manner. I can support the publication of this paper in Nature Communications.

In the revised version of the manuscript the Authors report additional experimental results (magnetic susceptibility and temperature dependence of resistivity and magnetoresistance), as well as provide additional discussion and new analysis of some results (critical field dependence on temperature). The sum of all results convincingly demonstrates an unconventional pressure-dependence of superconductivity in CsV_3Sb_5 concomitant with the suppression of the CDW and the intrinsic nature of the broadening of the superconducting transition between P1 and P2. I believe that these results can be of broad importance: on the one hand, they yield important information for the future identification of the possibly exotic SC and CDW orders in CsV_3Sb_5 ; on the other hand, they may help clarifying the origin of similar behaviors observed in other systems with competing orders (see below).

However, the scenario put forward by the authors (a new, nearly commensurate CDW with domain walls, where filamentary superconductivity occurs, appearing above P1) still appears to be not supported by the data. In particular:

(i) No features consistent with a second transition within the CDW phase (i.e. kinks, jumps) are observed between the CDW transition at T^* and T_c .

(ii) The magnetic susceptibility data (provided in the revision) clearly shows a strong broadening of the SC transition already between 0.49 and 0.66 GPa, i.e. before P1.

(iii) The onset of the diamagnetism coincides with the onset of resistivity suppression, while for quasi-1D filaments one would expect no diamagnetic signal (see also below).

(iv) As the SdH oscillations are suppressed by P1, non-CDW related changes in the Fermi surface can not be excluded, especially since the oscillations corresponding to the large pockets, presumably most relevant for SC, are not observed.

(v) Finally, the role of fluctuations and dimensionality is not assessed. CsV_3Sb_5 shows strong quasi-2D anisotropy and the coherence of superconductivity between the layers may play an important role. In particular, suppression of SC between P1 and P2 may indicate a disruption of the interlayer coupling which would lead to enhanced quasi-2D fluctuations broadening the transition. The Authors themselves draw a possible analogy with LBCO [PRL 99,067001 (2007)], where the suppression of SC occurs due to a decoupling of the layers by a commensurate density-wave order without requiring domain walls.

As the Authors point out themselves in their Reply, only other types of experiments (e.g. high-resolution x-ray scattering under pressure) may give conclusive evidence for their scenario. Therefore, I think that claims of a pressure-induced transition to a NCCDW with domain walls should be only mentioned in the discussion part and removed from figures and the rest of the text. In addition to that I list

below several suggestions to improve the manuscript:

(1) Regarding filamentary superconductivity: for SC confined to quasi-1D filaments one would not expect diamagnetic signal to develop at all (see [J. Phys. Chem. Solids Vol. 52, No. 6. pp. 761-767. 1991] or [PRB 103, 024502 (2021)] and [Fig. 3 in Annu. Rev. Condens. Matter Phys. 2019. 10:25-44] for examples). Strong anisotropy of the SC transition [PRB 85, 184513 (2012)] or dependence on the applied current value [Solid State Communications, 44, 12, 1539 (1982)] could also be expected. However, the Author's data shows instead a consistent onset of diamagnetism at the same temperature as the resistivity suppression and no anomalies mentioned above. This behavior rather suggests inhomogeneous superconductivity, where SC onsets gradually within the volume, rather than measure-zero filaments.

Consequently, the discussion of "filamentary" or inhomogeneous SC should be modified: in the current version the use of this notion is not explained in text and no references are given.

(2) Relation to works on other SC/density wave systems should be discussed more. In particular, a two-dome behavior has been observed previously in superconductors with competing density waves: JPSJ 82 033705 (2013), Nat. Comm. vol. 3, 943 (2012). Also, a nonmonotonic diamagnetic screening was reported Sci. Rep. 6:24068 (2016). While the control parameters in those studies are different (chemical substitution), the behavior is quite similar: close to the end point of the CDW phase SC is first enhanced on CDW suppression and then strongly suppressed, recovering only when moving further away from CDW phase. This suggests that the behavior reported by the Authors may be a general feature of SC/density wave competition.

(3) The details of the two-model fit of $H_{c2}(T)$ (formulas used and the resulting parameters) should be shown in the Supplementary; in particular, the dependence of the coherence lengths on pressure is a useful quantity and may be used to extract information about the scattering rates of individual bands.

(4) Can the superconducting volume fraction be extracted from the magnetization measurements? Does it reach 100 % or a close value for any pressure? This is an important point, since another Referee has pointed out a possibility of superconducting impurity phases.

(5) I think that the temperature dependence of magnetoresistance is an important result that should be emphasized (e.g. by discussing Sup. Fig. 5 somewhere around lines 126-133). It demonstrates that the linear MR is an intrinsic property of the CDW phase, as MR becomes quadratic when CDW is suppressed both by heating and pressure. A T-dependent MR curve outside of the CDW phase, showing that it is quadratic at all T, would make this statement even more convincing.

Note that in the previous version only low-T MR was shown, so it could have

been possible that linear MR is the property of the band structure not related to the CDW phase (the absence of changes in ρ_{300K} vs. P does not immediately imply the absence of change in MR).

(6) Related to the above, Fig. 4c misses an important feature of the data. Namely, the MR curves are clearly grouped in Supplementary Fig. 4 in three sets, roughly corresponding to regions $0 < P < P1$, $P1 < P < P2$ and $P2 < P$. Thus, if MR of a lower field, e.g. 3T, is plotted, one will be able to see that MR experiences two noticeable changes: around $P1$ and $P2$.

(7) Regarding QCP signatures in resistivity: the range of temperatures presented in Supp. Fig. 8 appears much smaller than the actual temperature range measured - I suggest to provide the ρ in full temperature range vs. T^2 in addition. Moreover, $\rho \sim T^2$ in a certain temperature range can be consistent with quantum criticality of a finite-Q order (such as CDW) in the presence of disorder - see e.g., PRB 51, 9253 (1995), PRL 82, 4280 (1999) - thus a larger temperature range can be of relevance.

Also, what appears more peculiar is the non- T^2 behavior at low pressures and temperatures - could the authors comment on that? In particular, for pressures where T_c is low, $\rho(T^2)$ dependence can be shown down to lowest temperatures, e.g. 2.5^2 K^2 for $P=0$ GPa. Indeed, data on a very similar compound KV_3Sb_5 [PHYSICAL REVIEW MATERIALS 3, 094407 (2019)] is fully consistent with T^2 behavior at low T , hence non- T^2 behavior for $P=0$ GPa appears anomalous.

Suggestions for Fig. 3:

(8) The coloring inside the CDW phase appears unrelated to data - perhaps MR% (see (6) above) or ρ can be used to provide a consistent and meaningful color scheme? Also, as is mentioned in the beginning of this Report, I suggest removing the NCCDW/domain cartoon as its presence is not supported by the data.

(9) The SC transition points (blue triangles/squares) in Fig. 3a appear not aligned with the brown-colored region, unlike Fig. 3b. Is this a plotting issue?

(10) The low-pressure region in Fig. 3b is rather crowded with points - I suggest to show an expanded view of the low-pressure region such that the correspondence between different measurement results could be better appreciated.

Wording/Typos:

Lines 19-24: it should be mentioned that SC increases dramatically after 1.1 GPa. Perhaps, this can be simply summarized by the "two-dome shape"

Line 25: "is linked" - > "is concomitant with" (a causal relation isn't demonstrated)

Line 41: "attentions" - > "attention"

Line 59: The results do not provide evidence of unconventional mechanisms of CDW and SC formation, but rather an unexpected (from the point of view of, e.g.,

Landau theory) competition between them.

Line 77: "much broad" - > "much broader"

Line 115,136,137,180: "emergence of a new CDW" state seems to imply a transition, but the data is also consistent with a crossover, so perhaps a "transformation (weak transition or crossover) of the CDW" is a better wording.

Line 129: "shows suddenly drops" - > "suddenly drops"

Line 147: "dramatically" - > "dramatic"

Line 173: "does" - > "do"

Reviewer #1 (Remarks to the Author):

The authors addressed the initial concerns I had about the manuscript. The revised discussion and presentation are now more in line with the data presented. I now support publication in Nature Communications.

Reply: We thank the reviewer for supporting publication in Nature Communications.

Reviewer #2 (Remarks to the Author):

In the revised manuscript the authors address my previous comments in a reasonable manner. I can support the publication of this paper in Nature Communications.

Reply: We thank the reviewer for supporting publication in Nature Communications.

Reviewer #4 (Remarks to the Author):

In the revised version of the manuscript the Authors report additional experimental results (magnetic susceptibility and temperature dependence of resistivity and magnetoresistance), as well as provide additional discussion and new analysis of some results (critical field dependence on temperature). The sum of all results convincingly demonstrates an unconventional pressure-dependence of superconductivity in CsV₃Sb₅ concomitant with the suppression of the CDW and the intrinsic nature of the broadening of the superconducting transition between P1 and P2. I believe that these results can be of broad importance: on the one hand, they yield important information for the future identification of the possibly exotic SC and CDW orders in CsV₃Sb₅; on the other hand, they may help clarifying the origin of similar behaviors observed in other systems with competing orders (see below).

However, the scenario put forward by the authors (a new, nearly commensurate CDW with domain walls, where filamentary superconductivity occurs, appearing above P1) still appears to be not supported by the data. In particular:

- (i) No features consistent with a second transition within the CDW phase (i.e. kinks, jumps) are observed between the CDW transition at T^* and T_c .
- (ii) The magnetic susceptibility data (provided in the revision) clearly shows a strong broadening of the SC transition already between 0.49 and 0.66 GPa, i.e. before P1.
- (iii) The onset of the diamagnetism coincides with the onset of resistivity suppression, while for quasi-1D filaments one would expect no diamagnetic signal (see also below).
- (iv) As the SdH oscillations are suppressed by P1, non-CDW related changes in the Fermi surface can not be excluded, especially since the oscillations corresponding to the large pockets, presumably most relevant for SC, are not observed.
- (v) Finally, the role of fluctuations and dimensionality is not assessed. CsV₃Sb₅ shows strong quasi-2D anisotropy and the coherence of superconductivity between the layers may play an important role. In particular, suppression of SC between

P1 and P2 may indicate a disruption of the interlayer coupling which would lead to enhanced quasi-2D fluctuations broadening the transition. The Authors themselves draw a possible analogy with LBCO [PRL 99,067001 (2007)], where the suppression of SC occurs due to a decoupling of the layers by a commensurate density-wave order without requiring domain walls.

As the Authors point out themselves in their Reply, only other types of experiments (e.g. high-resolution x-ray scattering under pressure) may give conclusive evidence for their scenario. Therefore, I think that claims of a pressure-induced transition to a NCCDW with domain walls should be only mentioned in the discussion part and removed from figures and the rest of the text. In addition to that I list below several suggestions to improve the manuscript:

Reply: We thank the reviewer's careful review and appreciate reviewer's constructive suggestions which will definitely improve the manuscript. We agree with the reviewer that the NCCDW with domain walls scenario is not fully confirmed yet. Thus, we take the referee's point and only mentioned it in the discussion part. Correspondingly, we have changed the related figures and discussions of the main text, and the modified text is marked in blue color in the revised manuscript.

(1) Regarding filamentary superconductivity: for SC confined to quasi-1D filaments one would not expect diamagnetic signal to develop at all (see [J. Phys. Chem. Solids Vol. 52, No. 6. pp. 761-767. 1991] or [PRB 103, 024502 (2021)] and [Fig. 3 in Annu. Rev. Condens. Matter Phys. 2019. 10:25-44] for examples). Strong anisotropy of the SC transition [PRB 85, 184513 (2012)] or dependence on the applied current value [Solid State Communications, 44, 12, 1539 (1982)] could also be expected. However, the Author's data shows instead a consistent onset of diamagnetism at the same temperature as the resistivity suppression and no anomalies mentioned above. This behavior rather suggests inhomogeneous superconductivity, where SC onsets gradually within the volume, rather than measure-zero filaments.

Consequently, the discussion of "filamentary" or inhomogenous SC should be modified: in the current version the use of this notion is not explained in text and no references are given.

Reply: We sincerely appreciate the reviewer's great suggestions. We agree with the referee that our resistivity and magnetic measurements are not significant to pin down a quasi-1D filamentary superconductivity. We modified our description as 'inhomogeneous superconductivity' as the referee suggested. However, such change will not alter our conclusions. We also added the discussion of the notion of inhomogeneous superconductivity (multi superconducting phases coexists) in the revised manuscript.

(2) Relation to works on other SC/density wave systems should be discussed more. In particular, a two-dome behavior has been observed previously in superconductors with competing density waves: JPSJ 82 033705 (2013), Nat. Comm. vol. 3, 943 (2012). Also, a nonmonotonic diamagnetic screening was reported Sci. Rep. 6:24068 (2016). While the control parameters in those studies are different (chemical substitution), the behavior is quite similar: close to the end point of the CDW phase SC is first enhanced on CDW suppression and then strongly

suppressed, recovering only when moving further away from CDW phase. This suggests that the behavior reported by the Authors may be a general feature of SC/density wave competition.

Reply: We thank the reviewer's suggestions. We added more discussion about the two-dome T_c behavior in the other superconductors with competing density waves. However, we would like to point out that our observation of two-dome behavior in CsV_3Sb_5 occurs WITHIN the CDW phase, which is different with the other superconductors (provided by the reviewer) where a second SC dome appears when the density wave order is completely suppressed. In the revised version, we have added 'Our discoveries may help to clarify the origin of similar behaviors observed in other systems with competing orders. In fact, two-dome T_c behavior were also observed in some superconductors³²⁻³⁴, which might be a general feature for superconductors with competing density waves. However, we would like to note that two-dome SC behavior in CsV_3Sb_5 occurs within the CDW phase, which is different with the other systems where a second SC dome appears when the density wave order is completely suppressed'.

(3) The details of the two-model fit of $H_{c2}(T)$ (formulas used and the resulting parameters) should be shown in the Supplementary; in particular, the dependence of the coherence lengths on pressure is a useful quantity and may be used to extract information about the scattering rates of individual bands.

Reply: We have added the details of the two-model fit of $H_{c2}(T)$ in the supplementary information. We also include the coherence lengths on different pressures in Supplemental Fig.9 (We also show the figure below for convenience) as the referee suggested. However, the scattering rates of individual bands are still difficult to be calculated due to the lacking information of electron mean-free path and Fermi velocity for each band at high pressure.

(4) Can the superconducting volume fraction be extracted from the magnetization

measurements? Does it reach 100 % or a close value for any pressure? This is an important point, since another Referee has pointed out a possibility of superconducting impurity phases.

Reply: Actually, the superconducting volume in our sample reaches 100% at ambient pressure, as evidenced both by magnetic susceptibility and heat capacity measurements (arxiv: 2102.10987 and arxiv: 2103.11796). In the high-pressure magnetic measurements, the superconducting volume fraction is calculated to be even larger than 100% (except for $P=0.72$ and 0.89) from our high-pressure magnetic measurements. In our high-pressure magnetic susceptibility experiment, magnetic field cannot be strictly along the *ab* plane because the sample is surrounded by liquid pressure medium. Therefore, the calculated superconducting volume fraction can be larger than 100% since the demagnetizing factor is not taken into consideration, especially for flake-like sample. On the other hand, demagnetizing factor do not need to be taken into account for the impurity phases because they are distributed uniformly in the sample (if exist), and the superconducting volume fraction cannot exceed 100%. Thus, the large diamagnetic signal could not relate to the impurity phases.

(5) I think that the temperature dependence of magnetoresistance is an important result that should be emphasized (e.g. by discussing Sup. Fig. 5 somewhere around lines 126-133). It demonstrates that the linear MR is an intrinsic property of the CDW phase, as MR becomes quadratic when CDW is suppressed both by heating and pressure. A T-dependent MR curve outside of the CDW phase, showing that it is quadratic at all T, would make this statement even more convincing. Note that in the previous version only low-T MR was shown, so it could have been possible that linear MR is the property of the band structure not related to the CDW phase (the absence of changes in ρ_{300K} vs. P does not immediately imply the absence of change in MR).

Reply: We appreciate the referee for pointing out this issue. We agree with the referee that temperature dependence of MR is an important result. We added 'Temperature dependence of MR at high pressure also evolves from "V" shape to "U" shape as shown in Supplementary Fig.5. These results indicate that the low-field linear MR is an intrinsic property of the CDW phase, since the MR exhibits quadratic temperature-dependence when the CDW is suppressed by both of heating and pressure.' in the revised manuscript as the referee suggested.

(6) Related to the above, Fig. 4c misses an important feature of the data. Namely, the MR curves are clearly grouped in Supplementary Fig. 4 in three sets, roughly corresponding to regions $0 < P < P_1$, $P_1 < P < P_2$ and $P_2 < P$. Thus, if MR of a lower field, e.g. 3T, is plotted, one will be able to see that MR experiences two noticeable changes: around P_1 and P_2 .

Reply: We thank the reviewer's great suggestions. We added the MR data at 3T in Fig.4 in the revised manuscript. The related description was also added in the revised manuscript.

(7) Regarding QCP signatures in resistivity: the range of temperatures presented in Supp. Fig. 8 appears much smaller than the actual temperature range measured - I suggest to provide the ρ in full temperature range vs. T^2 in addition. Moreover, $\rho \sim T^2$ in a certain temperature range can be consistent with quantum criticality of a finite-Q order (such as CDW) in the presence of disorder - see e.g., PRB 51, 9253

(1995), PRL 82, 4280 (1999) - thus a larger temperature range can be of relevance. Also, what appears more peculiar is the non- T^2 behavior at low pressures and temperatures - could the authors comment on that? In particular, for pressures where T_c is low, $\rho(T^2)$ dependence can be shown down to lowest temperatures, e.g. 2.5^2 K^2 for $P=0 \text{ GPa}$. Indeed, data on a very similar compound KV_3Sb_5 [PHYSICAL REVIEW MATERIALS 3, 094407 (2019)] is fully consistent with T^2 behavior at low T , hence non- T^2 behavior for $P=0 \text{ GPa}$ appears anomalous.

Reply: We thank the referee's great suggestion. We have shown ρ in a much larger temperature range in Supp. Fig.8 (We also show the figure below for convenience) as the referee suggested. Actually, the low-temperature resistivity follows T^2 behavior below 35 K above P_2 . It severely deviates from T^2 at higher temperature. The anomaly of resistivity with pressure between 1.64 and 1.84 GPa is related to the CDW transition. We should mention that the T^2 behavior can only be observed at low temperature even for a Fermi liquid system. At higher temperature, the electron-phonon scattering will make the resistivity deviate from the T^2 behavior. The low-field MR shows linear behavior in the CDW state, which possibly arises from the linear band crossings, can possibly make the resistivity deviate from the T^2 behavior. One of the remarkable differences between CsV_3Sb_5 and KV_3Sb_5 is that the former has very little (even no) vacancy. The vacancy in KV_3Sb_5 is much more significant which could lead to slightly different resistivity behavior. However, it is still an open question that needs to be clarified in future study.

Suggestions for Fig. 3:

(8) The coloring inside the CDW phase appears unrelated to data - perhaps MR% (see (6) above) or ρ can be used to provide a consistent and meaningful color scheme? Also, as is mentioned in the beginning of this Report, I suggest removing the NCCDW/domain cartoon as its presence is not supported by the data.

Reply: We have used the magnitude of MR (measured at 9T, 10K) as the color scheme in Fig.3 as the referee suggested. We have also removed the NCCDW/domain cartoon.

(9) The SC transition points (blue triangles/squares) in Fig. 3a appear not aligned with the brown-colored region, unlike Fig. 3b. Is this a plotting issue?

Reply: We thank the reviewer for point out this problem. We modified it in the revised manuscript.

(10) The low-pressure region in Fig. 3b is rather crowded with points - I suggest to show an expanded view of the low-pressure region such that the correspondence between different measurement results could be better appreciated.

Reply: We thank the reviewer's great suggestions. We expanded the view of the low-pressure region in Supplemental Fig.10 (We also show the figure below for convenience) in the revised manuscript.

Wording/Typos:

Lines 19-24: it should be mentioned that SC increases dramatically after 1.1 GPa. Perhaps, this can be simply summarized by the "two-dome shape"

Line 25: "is linked" -> "is concomitant with" (a causal relation isn't demonstrated)

Line 41: "attentions" -> "attention"

Line 59: The results do not provide evidence of unconventional mechanisms of CDW and SC formation, but rather an unexpected (from the point of view of, e.g., Landau theory) competition between them.

Line 77: "much broad" -> "much broader"

Line 115,136,137,180: "emergence of a new CDW" state seems to imply a transition, but the data is also consistent with a crossover, so perhaps a "transformation (weak transition or crossover) of the CDW" is a better wording.

Line 129: "shows suddenly drops" -> "suddenly drops"

Line 147: "dramatically" -> "dramatic"

Line 173: "does" -> "do"

Reply: We thank the referee for pointing out these typos. We made the related corrections in the revised manuscript.